# Epidemiology of Wilson’s Disease and Pathogenic Variants of the *ATP7B* Gene Leading to Diversified Protein Disfunctions

**DOI:** 10.3390/ijms25042402

**Published:** 2024-02-18

**Authors:** Elena Vasilievna Ovchinnikova, Mikhail Maksimovich Garbuz, Anna Aleksandrovna Ovchinnikova, Vadim Vladimirovich Kumeiko

**Affiliations:** 1Institute of Life Sciences and Biomedicine, School of Natural Sciences, Far Eastern Federal University, Vladivostok 690922, Russiagarbuzmihail.93@gmail.com (M.M.G.);; 2A.V. Zhirmunsky National Scientific Center of Marine Biology, Far Eastern Branch of Russian Academy of Sciences, Federal University, Vladivostok 690041, Russia

**Keywords:** Wilson’s disease (WD), copper transport ATPase, prevalence, genetic diagnosis

## Abstract

Wilson’s disease (WD) is an autosomal recessive disorder characterized by toxic accumulation of copper in the liver, brain, and other organs. The disease is caused by pathogenic variants in the *ATP7B* gene, which encodes a P-type copper transport ATPase. Diagnosing WD is associated with numerous difficulties due to the wide range of clinical manifestations and its unknown dependence on the physiological characteristics of the patient. This leads to a delay in the start of therapy and the subsequent deterioration of the patient’s condition. However, in recent years, molecular genetic testing of patients using next generation sequencing (NGS) has been gaining popularity. This immediately affected the detection speed of WD. If, previously, the frequency of this disease was estimated at 1:35,000–45,000 people, now, when conducting large molecular genetic studies, the frequency is calculated as 1:7026 people. This certainly points to the problem of identifying WD patients. This review provides an update on the performance of epidemiological studies of WD and describes normal physiological functions of the protein and diversified disfunctions depending on pathogenic variants of the *ATP7B* gene. Future prospects in the development of WD genetic diagnostics are also discussed.

## 1. Introduction

Wilson’s disease (WD) (OMIM277900) is a severe autosomal recessive disease caused by excessive accumulation of copper in the body and its toxic effects on various organs and systems. It has a wide range of onset (from 3 to 74 years) and is characterized by an unusually wide variability of symptoms, reflecting signs of damage to various organs and systems (mainly the liver and brain). Due to the wide variability of symptoms and difficulties in diagnosis, it is not always possible to recognize WD in the early stages. Therefore, WD has long been considered a rare hereditary disease with a fatal outcome [1]. The pathology was first described in 1883 by C. Westphal and A. Strumpell. Because of the similarity of the trembling hyperkinesia that occurs in such patients to similar trembling in patients with multiple sclerosis, they called it “pseudosclerosis.” The disease received its modern name—hepatolenticular degeneration in honor of Samuel Wilson. He, in 1912, having discovered degenerative changes in both the liver and the lenticular nuclei of the brain in patients with tremulous hyperkinesis, proposed calling the disease “progressive lenticular degeneration” [2].

In 1953, A.G. Bearn established the autosomal recessive mode of inheritance of the disease, and, in 1956, J.M. Walshe synthesized a copper-eliminating drug, D-penicillamine, from penicillin. In 1974, evidence was provided for various pathogenic variants in the *ATP7B* gene in the development of impaired biliary copper excretion in patients with WD [1]. According to official statistics, the prevalence of WD in the world ranges from 1 to 3 per 100,000 population: in Europe—between 1.2 and 2 per 100,000; in the USA—1 per 30,000 [2]; in Russia—1:166,600 [3].

Despite the rarity of detection, the contribution of this pathology to the disability of the young population reaches high levels and causes significant economic damage to healthcare. At the same time, an effective etiotropic therapy based on the use of copper-eliminating drugs (d-penicillamine, trientine, tetrathiomolybdate, unithiol, etc.) has been developed for WD. Therefore, taking into account that the number of cases contributes to the optimal calculation of the economic costs of healthcare for the prevention and treatment of the disease, an obstacle to solving this problem has always been the lack of clear criteria for its clinical and laboratory diagnosis. Patients with WD experience a wide range of symptoms related to various organs. The most common symptoms are liver dysfunction, neuropsychiatric disorders, Kayser–Fleischer rings on the cornea, and hemolysis caused by acute liver failure. Symptoms of WD can appear at any age. An accurate diagnosis is established after a thorough assessment of the clinical picture and genetic and biochemical tests, as well as after identifying a violation of copper metabolism. Biochemical tests may include determining the level of ceruloplasmin in the blood, studying the level of copper in 24-h urine, and a liver biopsy. However, even in this case, one should carefully evaluate the data and adjust the results taking into account the patient’s age, habits, and lifestyle and not exclude the possibility of similar diseases [4].

To identify WD, a careful analysis of the history of the patient and his relatives should be carried out since there may be atypical symptoms such as respiratory failure or a combination of WD with other diseases [5,6]. Particular attention should be paid to children with suspected WD. Among childhood symptoms, liver cirrhosis and acute liver failure are much more common, and they are also often asymptomatic [7]. In this regard, a comprehensive biochemical study and the use of genetic diagnostic methods are necessary. The most promising strategy for controlling the detection of WD is genetic screening of the country’s population and the creation of a huge registry of patients [8,9]. However, this is an expensive solution.

When identifying WD in children and adolescents, pharmacological therapy including chelating agents (e.g., penicillamine or trientine) and zinc salts is effective if the disease is diagnosed at an early stage and treated correctly. The transition period for children and adolescents is a difficult period, and, therefore, they may neglect therapy. Poor compliance is a serious problem and parents need to strictly monitor compliance with therapy to avoid worsening the child’s condition [10].

With the discovery of the *ATP7B* gene responsible for the molecular mechanism of WD, there was hope in determining the main criteria for diagnosing this pathology in terms of genetic studies. It was found that WD is caused by various pathogenic variants in the *ATP7B* gene (OMIM 606882), localized in the 13q14.3 region, which lead to disruption of intracellular transport of copper in hepatocytes, its incorporation into the ceruloplasmin molecule, and excretion of excess. In the process of research, it turned out that the frequency of heterozygous carriage of pathogenic variants in the *ATP7B* gene is 1 per 90, differing by a pronounced intrafamilial phenotypic differentiation of their combinations [11]. With such a frequency of heterozygous carriage of pathogenic variants, the annual registration of WD can reach 1 in 25,000–30,000, and, therefore, epidemiological indicators should vary and increase in different countries depending on a progress in clinical diagnostics [12]. This review highlights current findings from epidemiological studies of WD. The review also describes the functions and structure of the nonpathogenic *ATP7B* protein and the effect of pathogenic variants of the gene on the function and maturation of the protein.

## 2. Discussion

By the beginning of the new millennium, the number of patients with WD in the world reached 30 million [1]. Recent decades have seen an increase in epidemiological research. This is due to improved methods for diagnosing WD, increased awareness of this disease, and cheaper molecular genetic diagnostic methods.

To date, a group of scientists from the University of Alberta (Canada) has created an extensive open computer database of pathogenic variants, which has demonstrated the uneven distribution of WD in the world with a predominance in isolates—regions where closely related marriages are practiced due to limited population migration (Sardinia, Yemen, Iran, Jordan, and Northern India) [13,14,15], geographical isolation (Iceland) and due to ethnic assortativity (among Jews of Eastern European origin) [16] (Table 1).

By the 90s of the twentieth century, the frequency of detection of WD did not exceed 0.4:100,000 in Scotland [16], in Ireland—0.67:100,000 [15], in the USA—2:100,000 [26], and in Sardinia—2.9:100,000 [25]. Already in the first decades of the 21st century, its level increased significantly, reaching in the UK—14.2:100,000 [21], in Ireland—9.0:100,000 [15], in the USA—6.34:100,000 [26], in Sardinia—36.6:100,000 [13], in France, Brazil and Spain by 2016—stably kept within the range from 1.44:100,000 to 1.65:100,000 [22,23,24].

In some regions of the Russian Federation, despite official statistics on the low registration of WD in general (0.4:100,000 of the population), the incidence of the disease increased to: 1.5:100,000 [17], 4:100,000 [18], and 5.6–6.5:100,000 [19], respectively, in the Khabarovsk Territory, Udmurt Region, and Primorsky Territory.

In various European countries, the level of WD registration began to reach 1:7000–10,000, which significantly exceeds the data of previous years. The highest rates remained, among European countries, in Sardinia (36.6:100,000) [13] and Great Britain (14.2:100,000) [21], and among Asian countries in Korea (13:100,000) [27].

Over a short period of time in different countries from a large number of populations in patients with WD, more than 900 different *ATP7B* pathogenic variants were identified, and an unusual phenotypic differentiation was found in their combinations [29,30,31]. It turned out that the frequency of detection and the spectrum of pathogenic variants of the *ATP7B* gene differ significantly in different countries. At the same time, the level of prevalence of pathogenic variants in the *ATP7B* gene depends not only on the geographical location of the territory of residence of patients, but also on the ethnicity of the patient, and differences in indicators can also be traced back to representatives of the same population (Figure 1).

Copper-transporting ATPases belong to the P1 (CPX-, P1B-) subfamily of P-type ATPases. Members of this subfamily are involved in the transport of various transition metals (Cu, Ag, Cu2, Zn2, Ni2, Pb2) across cell membranes, in contrast to other P-type ATPases, which transport non-heavy metals (Na, K, Ca2, Mg2) or protons [31]. *ATP7B* normally has two functions in the liver. First, it transports copper to the Golgi apparatus for incorporation into ceruloplasmin and exports excess copper, binding the metal in vesicles for subsequent excretion into bile. The second function requires delivery of *ATP7B* from the Golgi apparatus to endocytic vesicles in response to increased intracellular copper concentrations [32,33].

The *ATP7B* structure includes several regions. The tail at the amino terminus contains six metal-binding domains (previously called copper-binding units), each of which has a characteristic CXXC amino acid motif. Eight transmembrane segments form a pore. The nucleotide binding site is located in the cytoplasm and consists of three domains: an N domain for binding ATP/ADP nucleotides, a P domain (phosphatase domain) where the invariant aspartate residue is transiently phosphorylated, and an A domain (activator) that facilitates the catalytic process. The carboxy-terminal region is partially disordered and contains a trileucine motif that regulates intracellular traffic [34] (Figure 2).

*ATP7B*-mediated copper transport involves several steps. First, *ATP7B* binds copper through its cytosolic N-terminal metal-binding domain and ATP through its nucleotide-binding domain. ATP is then hydrolyzed, and *ATP7B* is temporarily phosphorylated at the Asp1027 residue located in the P domain (catalytic phosphorylation). Subsequent dephosphorylation releases the energy required to transport copper across the membrane (transfer step) (Figure 3) [35].

Each of these steps can be affected by pathogenic variants that cause WD [31]. The effect may result in complete loss of *ATP7B* function if the mutated residues are critical for ATP or copper binding and/or conformational transitions during catalysis. Inactivation of *ATP7B* can also be partial if pathogenic variants reduce affinity for substrates, slow conformational transitions, or interfere with precise targeting of the protein to the Golgi apparatus or vesicles. The defective *ATP7B* accumulates and is deposited in the Golgi apparatus. Understanding phenotypic diversity in WD requires knowledge of how disease-causing pathogenic variants alter protein stability, activity, and cellular localization. Currently, such detailed information is not available for most pathogenic variants that cause WD [34].

*ATP7B* pathogenic variants can be divided into the following groups: (1) missense pathogenic variants, causing single amino acid substitutions in the protein sequence, (2) nonsense pathogenic variants, leading to the insertion of a stop codon and premature termination of translation, (3) frameshift pathogenic variants, usually caused by deletion or insertion of multiple nucleotides, and (4) splicing pathogenic variants and large gene rearrangements that result in gross modification or complete loss of the transcript. Pathogenic variants of types 2–4 often lead to the loss of the protein’s ability to perform its functions and the death of the patient in the early stages of his life, so the attention of researchers is focused on missense pathogenic variants [31]. Also, defects in the *ATP7B* gene may result from the accumulation of other metals (iron, manganese) in the nervous tissue of patients with WD. The most popular hypothesis for this effect is related to Cu2+ paramagnetism, since it is unknown what type of copper is deposited during WD [36].

There is a correlation between pathogenic variants in the *ATP7B* gene, phenotypic manifestations, and the course of the disease. Some *ATP7B* pathogenic variants prevent the protein from folding, as a result of which it does not enter the cytoplasm from the endoplasmic reticulum and is degraded. The most common example of such pathogenic variants is the missense pathogenic variant His1069Gln. In vitro studies have shown that this pathogenic variant makes it impossible for the NH domain of *ATP7B* to fold at physiological temperatures, causing its degradation in the endoplasmic reticulum [37]. This results in the protein being unable to reach the Golgi apparatus and, therefore, the site of copper excretion. With this pathogenic variant, patients have low levels of ceruloplasmin and high levels of free copper. Patients with the p.His1069Gln substitution show a moderate course of the disease with a predominant manifestation of neuropsychiatric symptoms at ages between 20–22 years [38,39]. It turns out that Kayser–Fleischer rings are more common in homozygous His1069Gln patients than in compound heterozygous individuals [40].

The retention of defective *ATP7B* in these organelles is due to the fact that, for the protein to enter the cytoplasm, it needs to receive the universal sorting signal AspLysTrpSerLeuLeuLeu, which is attached to the C-terminus of the protein. However, it was found that the addition of exogenous copper to the cell growth medium stabilizes the protein, allowing it to complete its planned migration to the Golgi apparatus and overcome the disease-causing phenotype. Theoretically, patients with this particular variant may be more sensitive to dietary copper deficiency [41].

Accordingly, the p.His1069Gln pathogenic variant prevents the protein from binding to AspLysTrpSerLeuLeuLeu, which leads to the inability of the *ATP7B* enzyme to reach the copper excretion site, causing copper deposition in cells and their toxic damage [42,43]. However, studies by Lalioti et al. demonstrated partial retention of copper transport function, possibly explaining the milder phenotypes associated with certain pathogenic variants.

Among the pathogenic variants that slow down the passage of defective *ATP7B* through the Golgi apparatus, the most common are p.Thr977Met in transmembrane domain 6 and p.Pro1352Ser/Leu/Arg in domains 7 and 8. These pathogenic variants correspond to the p.Thr994Ile and Pro1386Ser pathogenic variants in ATP7A, which cause severe Menkes disease. However, they have been proven to cause movement disorders and decreased mobility in patients. This once again shows the complexity and multifactorial nature of the manifestations of copper intoxication [44,45]. The p.Asn41Ser pathogenic variant leads to disruption of the signal sequence PheAlaPheAspAsnValGlyTyr, which is responsible for transport by basolateral endosomes, resulting in the accumulation of defective *ATP7B* in the basolateral plasma membrane of liver cells [42,46].

There is also an assumption about the role of other genes that influence the appearance of WD and change the clinical phenotype. These genes include *MTHFR* [4], *COMMD1* [47], *ATOX1* [48], *XIAP* [49], *PNPLA3* [50], and *DMT1* [51]. However, none of these genes have demonstrated significant diagnostic or prognostic value. In 2013, Coffey et al. conducted a study in the United Kingdom and found that only one gene, *ATP7B*, is responsible for the manifestation of WD [21].

It has been established that 25–30% of pathogenic variants lead to only partial inactivation of the *ATP7B* gene [35]. Pathogenic variants have been identified (pathogenic variant p.Ser653Tyr) that do not lead to disruption of copper transport, but also do not allow the protein to leave the Golgi complex [42]. With the p.Gly943Ser pathogenic variant, only copper excretion into bile is affected while its binding to ceruloplasmin remains normal [52]. Some pathogenic variants cause protein retention in the endoplasmic reticulum and failure of its transport to the Golgi complex, but this delay can be reversible [53,54].

The level of detection of the *ATP7B* gene pathogenic variant depends not only on the geographic location of the patients’ area of residence, but also on the ethnicity of the patient. At the same time, differences in indicators can be traced among representatives of the same population.

An example is the difference in the frequency of detection of the point pathogenic variant 3207C>A in exon 14, leading to the replacement of C with A in the p.His1069Gln codon, which leads to the replacement of the amino acid Histidine (His) with Glutamine (Gln). It was found that the p.His1069Gln pathogenic variant predominates on most chromosomes of WD patients in Europe and America, but among Slavs with WD living in Central and Western Europe, the allelic frequency of the p.His1069Gln pathogenic variant is 38% [27]. In the bordering countries of Eastern Europe, next to which the territory of the Russian Federation is located, the frequency of its detection reaches the highest values in Poland (72%), Lithuania (69.2%), the Czech Republic (57%), East Germany (63%), Belarus (61%), Latvia (52.2%) [51], and Ukraine (100%) (and, in Ukraine, 50% are cases of pathogenic variants in the homozygous state) [52] (Table 2).

As we move to the west and south of Europe, the level of detection of the p.His1069Gln pathogenic variant decreases its share: in Hungary—43% [57], in Serbia—38.4% [61], in Austria—34% [4], and in the UK—19% [21].

In the Russian sample, the share of the allelic frequency of the p.His1069Gln pathogenic variant, amounting to 31% to 50% in patients with WD, prevails in the Western regions of the country, where approximately 60% are detected in a homozygous state [55,63,64]. Moreover, in different ethnic groups, even in the same territory of the country, the proportion of the p.His1069Gln pathogenic variant differs. Thus, in the Republic of Bashkortostan, where, among 8 pathogenic variants identified in the *ATP7B* gene, the p.His1069Gln pathogenic variant is found with the highest frequency, its share among representatives of Russian origin is 55.5% [65], Tatar—38.9% [66], Bashkir—44.4% [67], and Chuvash—83.3% [68]. This suggests that, having “originated” in Eastern Europe, the p.His1069Gln pathogenic variant spread to the territory of Russia following migration flows. Less commonly, other pathogenic variants of the *ATP7B* gene are observed in our country, including those not described in other countries, among which two, previously not identified, are described by Karunas A.S. [63] and three by Bayazutdinova G.M. [55].

In other populations and territories of the world, other clusters of pathogenic variants are characteristic and are detected mainly in exons 2, 5, 8, 14, 15, 16, and 18 of the gene. Thus, in patients with WD living in different countries, the following predominate in the spectrum of pathogenic variants of the *ATP7B* gene options: p.Met645Arg—in Spain [23,63], p.Met769HisfsTer26—in Hong Kong and Taiwan [69], p.Pro992Leu and p.Arg778Leu—in other countries of East Asia and China [26,70], p.Ala803Thr and p.Arg778Leu—in Japan [26,70], p.Thr1232Pro—in Colombia [71], p.Arg969Gln, p.Leu936STOP, p.Ile1148Thr, and p.His1069Gln—in Greece [72], p.Ala1135GlnfsTer13—in Brazil and Venezuela [73], p.Pro992Leu, c.2977insA, and c.3031insA (in exons 18 and 13)—in Northern India, p.Cys271STOP—in Eastern India, p.Gly1061Glu and p.Cys271STOP—in South India [74], c.del441-427—in Sardinia [13], p.Ser744Pro and p.Glu1399Arg in the complete absence of European and Asian major pathogenic variants—in Pakistan and Saudi Arabia [75], and c.2299insC, p.Gly710Ser, p.Ala1135GlnfsTer13 (in exon 15), and p.Arg969Gln (in exon 13)—in Central, Eastern, and Northern Europe [21]. In 2012, a new pathogenic variant of the *ATP7B* gene in exon 9 was described in patients in Austria [76], Table 3 shows data on the frequency of pathogenic variants in the *ATP7B* gene, characteristic of different countries (according to Chang I., 2017) [71].

As molecular genetic research methods improved, new data emerged on the frequency of carriage of the pathological gene in representatives of various age groups.

In European countries: in Italy (Sardinia) in 2013, when conducting genetic studies in 1,672,404 representatives of different age categories, the carrier level of the *ATP7B* gene pathogenic variant was 1:22,707 [13]; in Poland in 2016, when examining 385,000 children aged 10–14 years, the carriage rate of the pathological gene reached 1:49,000 [83]; in England, during genetic screening of 75,000 newborns (from 1995 to 2009), the frequency of homozygous carriage of pathogenic variants of the *ATP7B* gene was found to range from 1:2500 to 1:7026 newborns [21]; in Korea in 2017, when examining 1090 patients with various somatic pathologies, carriage of the *ATP7B* gene pathogenic variant was found within 1:7561 [27]; and in China (Hong Kong), when examining 7,336,600 adolescents and young men from 2000 to 2016, the carriage rate of the *ATP7B* gene was 1:793 [12].

In Japan, in 2020, a genetic study of 1208 samples from national genetic databases, including the Human Genetic Variation Database (HGVD) and Japanese Multi Omics Reference Panel (jMorp), the prevalence of homozygous *ATP7B* gene was 12.1:100,000 and 19.6:100,000, respectively.

Such discrepancies between the frequency of carriage of the *ATP7B* gene and the official registration data for WD have given rise to doubts about the previously calculated prevalence of this disease. Typically, prevalence is calculated as the number of cases of the disease detected divided by the population size and multiplied by 100. Table 4 shows examples of such discrepancies in the prevalence of WD in different countries (according to Poujous A, 2018; Coffey AJ, 2013; Jang JY, 2017) [1,21,27] (according to Bayazutdinova G.M. 2019).

When using such concepts in Russia, where, according to official data, the number of patients with WD registered in the Russian register of orphan diseases for 2016 corresponds to 1:167,000, and the carrier frequency of a pathological gene pathogenic variant is 1:50, the estimated prevalence of WD should be 1:10,000 [1,61].

In Europe: among Polish children 2.04:100,000 [83], among British newborns—14.23–40:100,000 [21], in Asia—among Koreans—13.23:100,000 [27]. These figures significantly exceed official statistics on the prevalence of WD, based on clinical and biochemical markers of the disease [28].

The differences between the high level of carriage of the pathogenic variants of the *ATP7B* gene and the low registration of WD can be explained both by underdiagnosis, by the level of genetic research methods, and by the lack of agreement among geneticists in a unified approach to the analysis of the data obtained, since there is already evidence that the clinical manifestations of WD depend not only on the sensitivity of tissues to copper toxicity, but on the location and type of pathogenic variant of the *ATP7B* gene [84].

## 3. Conclusions

The analysis of indicators from epidemiological studies presented in the literature over recent decades has shown a manifold increase in the detection of WD relative to old indicators. However, despite the significant increase in the registration of cases of WD in the world and the high level of detection of the carriage of pathogenic variants of the *ATP7B* gene in representatives of all age groups, the problem of predicting the course of the disease remains open. In many countries, the spectrum of pathogenic variants of the *ATP7B* gene characteristic of the local population has not been established. Data on the occurrence of WD and the prevalence of pathogenic variants in regions can facilitate genetic diagnosis. Awareness of doctors in regions with high incidence can help speed up the diagnosis of this disease and more detailed study of WD. Currently, the most common tool for studying WD is the creation of NGS panels, however, this expensive method is not available for widespread use in poor countries or countries with large areas. Cheaper methods based on modifications of the PCR method, such as ARMS [84], may be suitable in solving this problem. However, in recent years, researchers have often suggested that the course of the disease and symptoms may also be influenced by mutations in introns, which affect the process of protein maturation, leading to its defectiveness. Research on this topic is currently ongoing [85]. The success of solving the problem of the prevalence of this disease in different countries will depend on the extent to which it is possible to universally introduce molecular genetic research methods and reduce their cost to increase accessibility. The results of studying the epidemiology of WD will then become the basis for the development of programs aimed at improving the health of the patients bearing diversified pathogenic variants of the *ATP7B* gene. Future development of low-cost genetic diagnostics and extended screening should enhance early detectability of the disease. This will facilitate progress in effective treatment of WD and reduce the degree of possible disability.

## Figures and Tables

**Figure 1 ijms-25-02402-f001:**
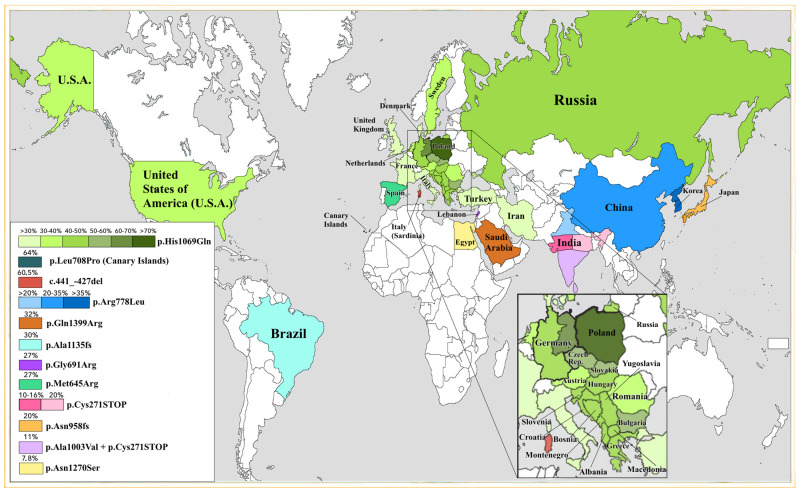
Distribution of the most common pathogenic variants of the *ATP7B* gene in different regions of the world. The table lists the most common mutations and their percentage among all identified cases.

**Figure 2 ijms-25-02402-f002:**
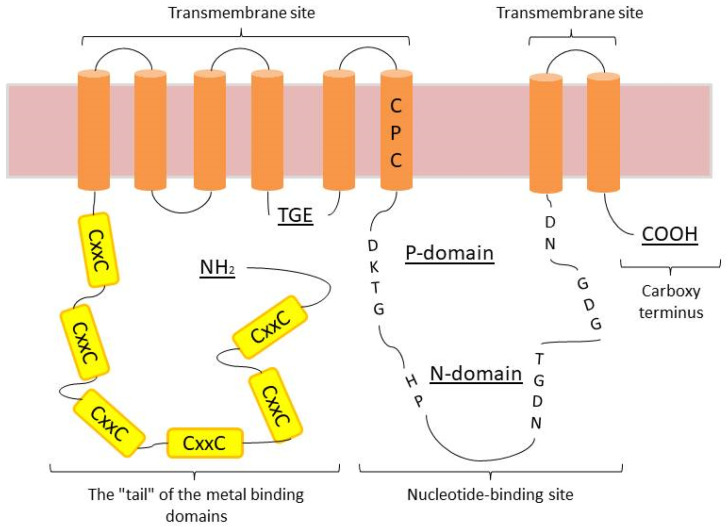
Structure and transmembrane organization of *ATP7B*.

**Figure 3 ijms-25-02402-f003:**
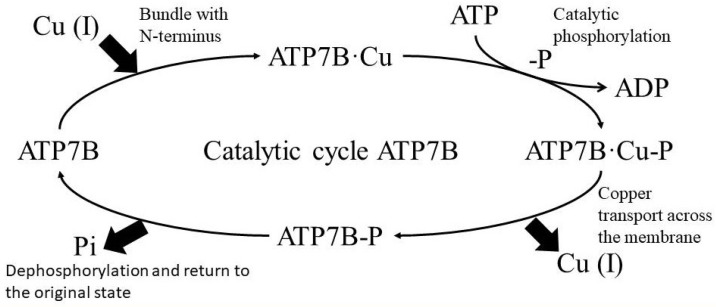
Scheme of the *ATP7B* catalytic cycle.

**Table 1 ijms-25-02402-t001:** Growth in the absolute number of patients and prevalence rates of WD in most countries of the world.

Country	Year of Research	WD Prevalence	Source
Russia	2014	0.23–2.6:100,000 (0.4:100,000)	[3]
RF: Khabarovsk Territory	2018	1.5:100,000	[17]
RF: Udmurt region	2018	4:100,000	[18]
RF: Primorsky Krai	2013	5.6–6.5:100,000	[19]
Scotland	1989		[20]
Ireland	1971	0.67:100,000	[15]
	2011	9.0:100,000	
Great Britain	2013	14.2:100,000	[21]
France	2013	1.5–1.6:100,000 for men, 1.44:100,000 for women	[22]
Spain (Gran Canaria)	2011–2013	8.08:100,000	[23]
Finland	2016	1.44:100,000	[24]
Italy (Sardinia)	1983	2.9:100,000	[25]
	2013	36.6:100,000	[13]
Brazil	2016	4.1:100,000	[25]
USA	2001	2:100,000	[26]
	2006–2011	6.4:100,000	[1]
Korea	2017	13:100,000	[27]
China (Hong Kong)	2000–2016	1.793:100,000	[28]

**Table 2 ijms-25-02402-t002:** Frequency of p.His1069Gln pathogenic variant detection in different countries.

Country	%	Source
Russian Federation	35–50	[55]
Austria	34	[4]
Bulgaria	58.8	[56]
Great Britain	19	[21]
Hungary	42.9	[57]
East Germany	63	[58]
Denmark	18	[59]
Italy	17.5	[25]
Poland	72	[31]
Romania	38.1	[60]
Serbia	38.4	[61]
USA	40.3	[26]
France	15	[22]
Czech	57	[62]

**Table 3 ijms-25-02402-t003:** Frequency of detection of major pathogenic variants in the *ATP7B* gene in different countries of the world.

Country	%	Pathogenic Variant	Source
Russia	35–50	p.His1069Gln (c.3207C<A)	[17,55]
Austria	34.1	p.His1069Gln (c.3207C>A)	[4]
	6.4	p.Gly710Ser (c.2128G>A)	
	3.4	p.Met769fs (c.2298_2299insC)	
Bulgaria	58.8	p.His1069Gln (c.3207C>A)	[56]
Canary Islands	64	p.Leu708Pro c.2123 (T>C)	[23]
Czech	57	p.His1069Gln (c.3207C>A)	[62]
Denmark	18	p.His1069Gln c.3207C>A	[59]
	16	p.Trp779STOP (c.2336G>A)	
France	15	p.His1069Gln (c.3207C>A)	[22]
West Germany	47.9	p.His1069Gln (c.3207C>A)	[4]
East Germany	63	p.His1069Gln (c.3207C>A)	[58]
Greece	35	p.His1069Gln (c.3207C>A)	[29]
	12	p.Arg969Gln (c.2906G>A)	
Hungary	42.9	p.His1069Gln (c.3207C>A)	[57]
Iceland	100	p.Tyr670STOP (c.2007_2013del)	[77]
Italy (continental)	17.5	p.His1069Gln (c.3207C>A)	[74]
	9	c.2530delA p.Val845fs	
Netherlands	33	p.His1069Gln (c.3207C>A)	[39]
Poland	72	p.His1069Gln (c.3207C>A)	[31]
	7.3	p.Ala1135GlnfsTer13 (c.3402delC)	
	3.7	p.Gln1351STOP (c.4051C>T)	
Romania	38.1	p.His1069Gln (c.3207C>A)	[60]
Sardinia	92	c.-441_-427del p.Met822fs (c. 2463delC)	[13,73]
Spain	27	p.Met645Arg (c.1934 T>G)	[23]
Sweden	38	p.His1069Gln (c.3207C>A)	[78]
Turkey	17.4	p.His1069Gln (c.3207C>A)	[79]
	5.3	p.Gly710Ser (c.2128G>A)	
Great Britain	19	p.His1069Gln (c.3207C>A)	[21]
	8	p.Met769Val (c.2305A>G)	
China	31	p.Arg778Leu (c.2332C>T)	[28]
	10	p.Pro992Leu (c.2975C>T)	[70]
	29	p.Arg778Leu (c.2332C>T)	[8]
Northern India	12	p.Ile1102Thr (c.3305 T>C)	[42]
	9	p.Pro992His (c.2975C>T)	
South India	11	p.Cys271STOP (c.813C>A)	[80]
	9	p.Pro768Leu (c.2303C>T)	[81]
Japan	17.95	p.Asn958fs (c.2871delC)	[26]
	16.7	p.Arg778Leu (c.2332C>T)	[70]
Korea	37.9	p.Arg778Leu (c.2332C>T)	[27]
	12.1	p.Asn1270Ser (c.3809A>G)	
Saudi Arabia	32	p.Gln1399Arg (c.4196A>G)	[75]
	16	p.Ser774Arg (c.2230 T>C)	
Iran	19	p.His1069Gln (c.3207C>A)	[14]
USA	40.3	p.His1069Gln (c.3207C>A)	[26]
	1.9	p.Asn1270Ser (c.3809A>G)	[26]
	1.9	p.Gly1266Arg (c.3796G>A)	
Brazil	37.1	p.His1069Gln (c.3207C>A)	[25]
	11.4	p.Ala1135GlnfsTer13 (c.3402delC)	
Venezuela	26.9	p.Ala1135GlnfsTer13 (c.3402delC)	[82]
	9.6	p.Gly691Arg (c.2071G>A)	

**Table 4 ijms-25-02402-t004:** Incidence of actual and old prevalence of WD in different countries.

Country	Actual Prevalence of WD	Old Prevalence of WD	Carriage Status of WD According to Whole Exome Sequencing Data
Great Britain	1:30,000	1:7026	1:25
France	1:67,000	1:9000	1:31
Korea			1:53
Russia	1:167,000	1:10,000	1:43

## Data Availability

Not applicable.

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
