# Peer review of "Epidemiology of Wilson’s Disease and Pathogenic Variants of the ATP7B Gene Leading to Diversified Protein Disfunctions"

_ijms, 2024, doi:10.3390/ijms25042402_

Round 1

Reviewer 1 Report

Comments and Suggestions for Authors

The manuscript contains a valuable and interesting summary and supplementation of current data on the occurrence of WD/disease carriage. Proposals for ways to simplify and reduce the cost of diagnosis in order to increase its availability are discussed very briefly. The authors should write something more on this topic or remove part of the introduction.

This article did not highlight (or did so very briefly) the problems associated with the failure to diagnose the disease, and it did not make a critical comparison of the detection of Wilson's disease by biochemical methods - therefore this information should be removed from the manuscript before publication. The sentence regarding forecasting the increase in the spread of WD in the population requires clarification - because in fact, the increase in the incidence of WD in the following years in various populations does not result from the spread of the disease but from progress in diagnostics and increased availability of genetic tests.

The manuscript also does not describe strategies to further improve diagnosis and develop personalized treatment for Wilson's disease. Therefore, this fragment of text should be removed from the introduction.

Also, the sentence in which the authors write about the mutation detection level (line 104) should be clarified. In fact, the level of detection of mutations in the ATP7B gene depends primarily on the availability of medical services and specialized diagnostic tests: biochemical and genetic. Geographic location and ethnicity are related to the frequency of specific ATP7B mutations, although this may vary among members of the same population.

 Figure 1 - it is not clear what is shown in the figure. What do the numbers indicated in the legend mean? (>30, 30-40, etc.)

The legend uses asterisks to symbolize some mutations, but it is not clear what they mean

 Line 160: Please complete the sentence with information in which organelles the retention of defective ATP7B occurs.

 Line 185-186: They have been proven to cause movement disorders and decreased sensitivity. Please explain what sensitivity you mean.

 Line 310 and beyond: It is difficult to expect that the results of epidemiological research will influence the development of programs aimed at improving the health of the world's population, because improving the health of the world's population requires primarily research on lifestyle diseases, not rare diseases.

It is also difficult to expect that new protocols for providing medical and social care will be developed, as these protocols are well developed in most countries, and the fact that WD is sometimes underdiagnosed is mainly due to limited access to specialist tests in some countries.

It is also difficult to agree with the statement that developing such protocols will slow down the rate of disease progression, because it is not an infectious disease and does not spread quickly. Its prevalence may be increasing in some populations, especially those with offspring from related people. These pieces of text need to be removed or written more correctly.

The title should be changed to better reflect the content of the work. The work concerns primarily the prevalence of various mutations in various populations around the world, various mechanisms of the impact of mutations on the defect in copper metabolism and the increase in mutation detection, which translates into an increase in the estimated occurrence of mutations and WD in populations.

Comments on the Quality of English Language

Improving the English language may improve the readability and understandability of the article.

Author Response

Dear reviewer, we thank you for reading our article, pointing out the shortcomings and helping us improve the quality of the material we wrote. Thank you for the work you've done.

Thank you so much for your recommendations, comments and observations. We have made your corrections to the text, they are marked in red. Thank you for your invaluable recommendations on the drawing and the title of the article.

Reviewer 2 Report

Comments and Suggestions for Authors

The review offers a comprehensive examination of Wilson's disease (WD), encompassing its historical context, genetic foundation, clinical manifestations, prevalence, and pathogenic variants within the ATP7B gene. In the discussion section, genetic aspects of WD are explored, elucidating the ATP7B gene's structure, various variants, and their impacts on protein function.

However, some clarifications are needed. There are inconsistencies in the presentation of statistical data. While the global prevalence of WD is stated as 1 to 3 per 100,000, regional rates vary (e.g., Europe—1.2 to 2 per 100,000). Yet, the prevalence in the USA is cited as 1 per 30,000, which appears inconsistent with the earlier global range. Ensure that all statistics and claims are substantiated with appropriate citations to enhance the review's credibility. References like "30 million people with WD by the beginning of the new millennium" seem excessively high and should be verified for accuracy.

In terms of minor revisions, the terms "actual" and "calculated" prevalence are introduced without clear definitions, causing confusion. A more precise explanation of these terms is necessary.

The review repeats certain information, such as details about the p.His1069Gln variant, in different sections. Streamlining the content and avoiding redundancy would enhance the review's clarity and flow.

Following the new nomenclature, "mutation" must be replaced with "pathogenic variants."

By addressing these points, the review can achieve greater cohesion, accuracy, and reader-friendliness.

Comments on the Quality of English Language

Grammatical errors and difficult phrasing are prevalent throughout the text. For example, in the conclusion section, "in underdiagnosis" should be corrected to "underdiagnosis," and other grammatical improvements are required for better readability.

Author Response

Dear reviewer, we thank you for reading our article, pointing out the shortcomings and helping us improve the quality of the material we wrote. Thank you so much for the work you've done.

Thank you so much for your recommendations, comments and observations. We have made your corrections to the text, they are marked in red.

Regarding your comments:

1) However, some clarifications are needed. There are inconsistencies in the presentation of statistical data. While the global prevalence of WD is stated as 1 to 3 per 100,000, regional rates vary (e.g., Europe—1.2 to 2 per 100,000). Yet, the prevalence in the USA is cited as 1 per 30,000, which appears inconsistent with the earlier global range. Ensure that all statistics and claims are substantiated with appropriate citations to enhance the review's credibility. References like "30 million people with WD by the beginning of the new millennium" seem excessively high and should be verified for accuracy.

1) thank you for pointing out this point, we double-checked and corrected the data that was incorrect. They are reliable and are referenced by other works.

2) In terms of minor revisions, the terms "actual" and "calculated" prevalence are introduced without clear definitions, causing confusion. A more precise explanation of these terms is necessary.

2) Thank you for pointing out this incorrectness, we have corrected this in the text into more understandable terms.

3) The review repeats certain information, such as details about the p.His1069Gln variant, in different sections. Streamlining the content and avoiding redundancy would enhance the review's clarity and flow.

3) We have corrected the problem you mentioned and restructured the text.

4) Following the new nomenclature, "mutation" must be replaced with "pathogenic variants."

4) Thank you for this information, we have corrected this term in the text.

By addressing these points, the review can achieve greater cohesion, accuracy, and reader-friendliness.

Reviewer 3 Report

Comments and Suggestions for Authors

Dear Authors,
Thank you for an interesting article, I have a few comments:
1. The authors attempted to describe the future direction of Wilson's disease diagnosis. Throughout the work, there is no reference to the current diagnostic recommendations for WD, e.g. for children from 2018. Although the authors focused solely on the genetic aspect, I understand that they also will describe current/other diagnostic methods like the presence of ceruloplasmin in blood serum, copper in 24-hour urine collection, and liver biopsy or eye examinations of WD patients. In the introduction, this should be mentioned.
2. There is no information on the differences in the symptoms of WD in children and adults. What prompts medical doctors to undertake patient diagnostics?
3. The bibliography needs to be reviewed and supplemented with current publications.

Author Response

Dear reviewer, we thank you for reading our article, pointing out the shortcomings and helping us improve the quality of the material we wrote. Thank you so much for the work you've done.

Thank you so much for your recommendations, comments and observations. We have made your corrections to the text, they are marked in red.

Regarding your comments:

1. The authors attempted to describe the future direction of Wilson's disease diagnosis. Throughout the work, there is no reference to the current diagnostic recommendations for WD, e.g. for children from 2018. Although the authors focused solely on the genetic aspect, I understand that they also will describe current/other diagnostic methods like the presence of ceruloplasmin in blood serum, copper in 24-hour urine collection, and liver biopsy or eye examinations of WD patients. In the introduction, this should be mentioned.

1) Thank you for your comment, we have updated the text in accordance with your corrections.

2. There is no information on the differences in the symptoms of WD in children and adults. What prompts medical doctors to undertake patient diagnostics?

2) Thank you for your comment, but we did not want to delve into the childhood symptoms of Wilson's disease since our team does not have specialists in this field. In this regard, we do not consider ourselves competent in this topic and are afraid that we could make many mistakes.

3. The bibliography needs to be reviewed and supplemented with current publications.

3) Thank you for your note, we tried to supplement the bibliographical list with more recent articles, but, unfortunately, this is not possible for all countries.

Round 2

Reviewer 3 Report

Comments and Suggestions for Authors

Reviewer response to Authors note:

 Dear reviewer, we thank you for reading our article, pointing out the shortcomings and helping us improve the quality of the material we wrote. Thank you so much for the work you've done.

 Thank you so much for your recommendations, comments and observations. We have made your corrections to the text, they are marked in red.

I'm grateful you appreciate my efforts. Regrettably, I still possess some suggestions for enhancement.

Regarding your comments:

 1. The authors attempted to describe the future direction of Wilson's disease diagnosis. Throughout the work, there is no reference to the current diagnostic recommendations for WD, e.g. for children from 2018. Although the authors focused solely on the genetic aspect, I understand that they also will describe current/other diagnostic methods like the presence of ceruloplasmin in blood serum, copper in 24-hour urine collection, and liver biopsy or eye examinations of WD patients. In the introduction, this should be mentioned.

1) Thank you for your comment, we have updated the text in accordance with your corrections.

2. There is no information on the differences in the symptoms of WD in children and adults. What prompts medical doctors to undertake patient diagnostics?

2) Thank you for your comment, but we did not want to delve into the childhood symptoms of Wilson's disease since our team does not have specialists in this field. In this regard, we do not consider ourselves competent in this topic and are afraid that we could make many mistakes.

 The Reviewer may find it challenging to comprehend the explanation that there is a lack of specialists in a particular field among the Authors in a given group. So why did the Authors decide to write this article on this topic?

There are many publications on this subject. I kindly ask you to read the publications:

Žigrai M, Vyskočil M, Tóthová A, Vereš P, Bluska P, Valkovič P. Front Med (Lausanne). Late-Onset Wilson's Disease. 2020 Feb 6;7:26. doi: 10.3389/fmed.2020.00026. eCollection 2020. PMID: 32118011

Zhang S, Yang W, Li X, Pei P, Dong T, Yang Y, Zhang J. Transl Neurodegener. Clinical and genetic characterization of a large cohort of patients with Wilson's disease in China.2022 Feb 28;11(1):13. doi: 10.1186/s40035-022-00287-0.

Sukezaki A, Chu PS, Shinoda M, Hibi T, Taniki N, Yoshida A, Kawaida M, Hori S, Morikawa R, Kurokouchi A, Wakino S, Kameyama K, Obara H, Kitagawa Y, Saito H, Kanai T, Nakamoto N. Clin J Gastroenterol. Late-onset acute liver failure due to Wilson's disease managed by plasmapheresis and hemodiafiltration successfully serving as a bridge for deceased donor liver transplantation: a case report and literature review 2020 Dec;13(6):1239-1246. doi: 10.1007/s12328-020-01175-8. Epub 2020 Jul 8. PMID: 32643122 Review.

Nilles C, Obadia MA, Sobesky R, Dumortier J, Guillaud O, Laurencin C, Moreau C, Vanlemmens C, Ory-Magne F, de Ledinghen V, Bardou-Jacquet E, Fluchère F, Collet C, Oussedik-Djebrani N, Woimant F, Poujois A. Mov Disord. Diagnosis and Outcomes of Late-Onset Wilson's Disease: A National Registry-Based Study. 2023 Feb;38(2):321-332. doi: 10.1002/mds.29292. Epub 2022 Dec 27. PMID: 36573661

Socha P, Czlonkowska A, Janczyk W, Litwin T. Wilson's disease- management and long term outcomes. Best Pract Res Clin Gastroenterol. 2022 Feb-Mar;56-57:101768. doi: 10.1016/j.bpg.2021.101768. Epub 2021 Oct 12.

Socha P, Janczyk W, Dhawan A, Baumann U, D'Antiga L, Tanner S, Iorio R, Vajro P, Houwen R, Fischler B, Dezsofi A, Hadzic N, Hierro L, Jahnel J, McLin V, Nobili V, Smets F, Verkade HJ, Debray D. Wilson's Disease in Children: A Position Paper by the Hepatology Committee of the European Society for Paediatric Gastroenterology, Hepatology and Nutrition. J Pediatr Gastroenterol Nutr. 2018 Feb;66(2):334-344. doi: 10.1097/MPG.0000000000001787.

3. The bibliography needs to be reviewed and supplemented with current publications.

3) Thank you for your note, we tried to supplement the bibliographical list with more recent articles, but, unfortunately, this is not possible for all countries.

The Reviewer considers this explanation inconsistent with the principles of publishing in scientific journals. Thanks to the Internet, you can always contact the Authors and ask for a PDF version of a given article. I don't know anyone who wouldn't respond positively to such a request. Therefore, I repeat my request to update the list of publications.

Author Response

Dear reviewer, we thank you for your work and careful review of our manuscript. We apologize for the inconvenience caused and also thank you for your patience. All fixes are marked in red.
Regarding your comments:

2) We read the publications that you recommended to us and based on them we wrote several paragraphs in the introduction. Thank you for your advice!

3) We reviewed the bibliography and changed it. Thank you for this note!

We hope that this version of the manuscript will satisfy you! Sincerely, the authors of the manuscript

Round 3

Reviewer 3 Report

Comments and Suggestions for Authors

Thank you for the corrections. One small note - what meens "BV" in line 73?